# The Variation of Selected Physiological Parameters in Elm Leaves (*Ulmus glabra* Huds.) Infested by Gall Inducing Aphids

**DOI:** 10.3390/plants11030244

**Published:** 2022-01-18

**Authors:** Katarzyna Kmieć, Izabela Kot, Katarzyna Rubinowska, Edyta Górska-Drabik, Katarzyna Golan, Hubert Sytykiewicz

**Affiliations:** 1Department of Plant Protection, University of Life Sciences in Lublin, Leszczyńskiego 7, 20-069 Lublin, Poland; katarzyna.kmiec@up.lublin.pl (K.K.); edyta.drabik@up.lublin.pl (E.G.-D.); katarzyna.golan@up.lublin.pl (K.G.); 2Department of Botany and Plant Physiology, University of Life Sciences in Lublin, Akademicka 15, 20-950 Lublin, Poland; katarzyna.rubinowska@up.lublin.pl; 3Institute of Biological Sciences, Siedlce University of Natural Sciences and Humanities, Prusa 12, 08-110 Siedlce, Poland; hubert.sytykiewicz@uph.edu.pl

**Keywords:** antioxidant enzymes, biotic stress, *Colopha compressa*, *Eriosoma ulmi*, *Tetraneura ulmi*

## Abstract

Three aphid species, *Eriosoma ulmi* (L.), *Colopha compressa* (Koch) and *Tetraneura ulmi* (L.) induce distinct gall morphotypes on *Ulmus glabra* Huds.; opened and closed galls. Because the trophic relationship of aphids and their galls shows that throughout the gall formation aphids can elicit multiple physiological regulations, we evaluated the changes of hydrogen peroxide content (H_2_O_2_), cytoplasmic membrane condition, expressed as electrolyte leakage (E_L_) and concentration of thiobarbituric acid reactive substances (TBARS), as well as, the activity of catalase (CAT), guaiacol peroxidase (GPX) and ascorbate peroxidase (APX) in gall tissues, as well as, in damaged and undamaged parts of galled leaves. All aphid species increased E_L_ from gall tissues and significantly upregulated APX activity in galls and galled leaves. Alterations in H_2_O_2_ and TBARS concentrations, as well as GPX and CAT activities, were aphid- and tissue-dependent. The development of pseudo- and closed galls on elm leaves did not have a clear effect on the direction and intensity of the host plant’s physiological response. The different modes of changes in H_2_O_2_, TBARS, CAT and GPX were found in true galls of *C. compressa* and *T. ulmi*. Generally, physiological alterations in new plant tissues were quite different compared to other tissues and could be considered beneficial to galling aphids.

## 1. Introduction

Host plants and insects interact at various levels, and gall formation is seen as a unique and extreme form of relationship. Insects as ‘gall-inducers’ change active differentiation and growth of plant tissues, manifested as remodeling of host anatomy and metabolism [1,2,3]. The development of plant galls depends on insect activity because each species uses a specific galling site and induces galls of different shapes and sizes [4,5,6]. *Colopha compressa* (Koch), *Eriosoma ulmi* (L.) and *Tetraneura ulmi* (L.) aphids have complex life cycles, with alternating sexual and parthenogenetic generations. In the spring, they induce galls on the leaf surfaces of various elm (*Ulmus* spp.) species as the primary host. Gall is induced only by first instar larva of neonate fundatrix hatching from an overwintered egg. When the gall is fully grown, the gall founder gives birth to offspring that develop into winged emigrant aphids that leave the gall and fly to the secondary host. Eggs are deposited on the trunk of the primary host by apterous sexual females, which are produced by alate migrants returning from the secondary host [7,8,9]. *T. ulmi* induces bean-shaped, stalked galls with a green, smooth surface. They are mostly formed on the apical part of the leaf blade. A single gall is initiated by a single fundatrix; however, many aphids can start the galling process on the same leaf. During gall formation, specific leaf distortions are observed [7,10]. Pocket galls of *C. compressa* are formed by a single fundatrix near the midrib. A single gall is laterally compressed, yellowish with a red tint, and looks like a cockscomb. It is situated mostly on the basal part of the leaf lamina and visible discoloration of leaf blades below the gall is observed [11,12]. Open, leaf-roll pseudogalls are induced usually by one, sporadically even several young *E. ulmi* fundatrices. Due to aphid feeding, the lateral edge of the leaf blade rolls up downwards, twisting, and blistering, thereby gradually forming a pseudogall [8,9].

The induction and development of galls is a biotic factor affecting plant condition and exposing host tissues to oxidative stress [1,13,14,15,16,17]. It is well known that host plants can initiate signal transduction in response to insect feeding and activate related physiological and biochemical reactions. Reactive oxygen species (ROS) are molecules of defense signaling pathways with known involvement in the activation of plant response to aphid attack [18,19]. Hydrogen peroxide (H_2_O_2_), as one of ROS, is relatively stable, mildly reactive and electrically neutral. It is able to pass through cell membranes and reach cell locations distant from the site of its formation. This ROS is important for signaling in plant growth and developmental processes as well as in reaction to biotic and abiotic stresses along with programmed cell death (PCD) [20]. The consequence of ROS overproduction is damage to proteins, DNA and lipids that may result in loss of function and formation of cytotoxic, low molecular weight degradation products [18,21,22]. ROS interfere with signaling pathways, thereby leading to the scavenging and detoxification of free radicals and other intermediates through antioxidant systems. Plants can control generated ROS by a set of antioxidants, for example, antioxidant enzymes, including catalase (CAT), which removes hydrogen peroxide by converting it into water and oxygen, and several peroxidases, which can also reduce H_2_O_2_ [20,23]. Guaiacol peroxidase (GPX) is a member of a large multigenic heme-containing enzyme family that controls ROS generation when plants are challenged with various stressors [23]. It also oxidizes a variety of phenolic compounds and participates in a broad range of physiological processes, like auxin catabolism, lignification and degradation of the cell wall [24]. H_2_O_2_ accumulation in plant cells is also controlled by ascorbate peroxidase (APX,), a key enzyme of the ascorbate-glutathione (ASA-GSH) cycle. APX is abundantly present in plant cells, and its isoforms acquire a significantly higher affinity for H_2_O_2_ than CAT [20], but the knowledge regarding the APX activity in plants challenged by insects is limited [25].

Aphids secrete during feeding a proteinaceous salivary sheath that lines the stylet path, as well as watery saliva containing numerous enzymes, such as oxidases, pectinases, or cellulases [26]. However, the ability of gall-inducing aphids to alter indirect plant defenses and the distribution of defensive compounds are poorly understood [4,11,12,15,17,27,28,29,30,31,32,33]. Therefore, to clarify the physiological changes in galls which can notable variability depending on gall-inducing species, host plant, or feeding period, we analyzed the H_2_O_2_ concentration, cytoplasmic membrane condition, and changes in the activity of catalase and peroxidases, such as GPX and APX. We attempted to determine changes in different parts in different parts of *Ulmus glabra* Huds. leaves galled by *E. ulmi*, *C. compressa* and *T. ulmi*. A similar pattern of host plant physiological reactions caused by *C. compressa* and *T. ulmi* feeding was expected, as both aphid species stimulate plant tissue to produce true (closed) galls on the upper side of the leaf blade, as opposed to *E. ulmi*, which induces pseudogalls.

## 2. Material and Methods

### 2.1. Plant Material and Samplings

The research was carried out on *Ulmus glabra* Huds. trees which are part of urban green areas of Lublin, Poland (51.24° N, 22.57° S). The galling activity of *C. compressa*, *E. ulmi* and *T. ulmi* is not synchronized, because fundatrices hatch from eggs at a different time. Thus, leaves galled by a particular aphid species and corresponding intact leaves were analyzed separately according to developmental differences of galls. Samples were taken when the galls were fully developed with a fundatrix and its offspring (2nd and 3rd stage) feeding inside. One sample consisted of 20 leaves with galls (for each aphid species) taken from 3–4 different trees within hand’s reach. Phenologically similar intact leaves situated on the shoots without galls were taken as control. Galled and intact leaves were detached from the same trees with scissors and kept in plastic bags. In the laboratory, within 1 h after collection, galls and pseudogalls were cut off from the leaves using a scalpel, and the aphids were removed by a soft brush. Parts of the leaf blade with visible damage were separated. For *C. compressa* and *T. ulmi* plant material was categorized as four combinations of the experiment: (1) control (intact) leaves, (2) undamaged parts of the lamina (without visible discoloration and corrugation) of galled leaves, (3) damaged parts of galled leaves, (4) galls alone. Three combinations were applied for *E. ulmi*, namely (1) intact leaves, (2) undamaged portions of galled leaves and (3) pseudogalls. 

### 2.2. Measurement of Hydrogen Peroxide, Lipid Peroxidation and Electrolyte Leakage

Hydrogen peroxide (H_2_O_2_) content was estimated by forming a titanium–hydroperoxide complex [34]. Fresh plant material (0.5 g) was ground in 3 mL of phosphate buffer (50 mM, pH 6.5) at 4 °C, the mixture was centrifuged at 6000× *g* for 25 min. Next, 1.5 cm^3^ of the supernatant was added to 0.5 cm^3^ titanium sulfate in 20% H_2_SO_4_ (*v*/*v*) and centrifuged at 6000× *g* for 15 min. The absorbance of the supernatant was measured at 410 nm against a blank reagent with a Cecil CE 9500 spectrophotometer (Cecil Instruments, Cambridge England). The H_2_O_2_ concentration in the sample was calculated using the molar absorbance coefficient 0.28 μM^−1^ cm^−1^, and are expressed as nanomoles per 1 g fresh weight.

The membrane lipid peroxidation was assessed by determining thiobarbituric acid reactive substances (TBARS) content according to Heath and Packer [35]. Crushed plant material (0.2 g) was homogenized in 0.1 M potassium phosphate buffer, pH 7.0, and centrifuged at 12,000× *g* for 20 min. Next, 0.5 cm^3^ of the homogenate was added to 2 cm^3^ 20% trichloroacetic acid (TCA) containing 0.5% thiobarbituric acid (TBA) incubated in a water bath at 95 °C for 30 min, quickly cooled and centrifuged at 10,000× *g* for 10 min. The absorbance was measured at 532 and 600 nm with a spectrophotometer mentioned above. The TBARS concentration in the samples was calculated using the molar absorbance coefficient (155 nM^−1^ cm^−1^). Results are presented as nanomoles per 1 g fresh weight.

Electrolyte leakage (E_L_) was measured with an Elmetron CC-317 microcomputer conductometer. Ten leaf/gall rings (9 mm diameter) were cut using a cork borer from each sample. The plant material was placed in test tubes containing 20 cm^3^ distilled deionized water. The tubes were incubated on the rotary shaker for 24 h at room temperature, and the initial electrical conductivity (K1) was determined. Then, the samples were autoclaved at 100 °C for 15 min, and after 24 h of shaking the final conductivity of the solution was measured (K2). Electrolyte leakage was calculated using the formula: E_L_ (%) = (K1/K2) × 100 [36].

### 2.3. Assay of Antioxidant Enzymes Activities

Fresh leaf material (0.2 g) was homogenized with 10 cm^3^ of 50 mM phosphate buffer, pH 7.0 containing 0.2 M EDTA (ethylenediaminetetraacetic acid) and 2% PVP (polyvinylpyrrolidone). The homogenate was centrifuged at 10,000× *g* for 15 min at 4 °C, and obtained supernatant was used for enzyme analysis. Absorbance readings were performed using a Cecil CE 9500 spectrophotometer.

Catalase (CAT; EC 1.11.1.6) activity was measured by Chance and Meahly [37] method with Wiloch et al. [38] modification. The reaction mixture contained 2 cm^3^ of 50 mM potassium phosphate buffer pH 7.0, 0.2 cm^3^ of H_2_O_2_ and 0.1 cm^3^ of supernatant. The extinction was measured for 3 min reading at the initial and final stage at 240 nm. Catalase activity was determined using the absorbance coefficient 0.036 mM^−1^ cm^−1^. The results were converted to catalase activity per fresh weight, expressed as U mg^−1^ fresh weight. The activity of peroxidase towards guaiacol (GPX; EC 1.11.1.7) was assayed as described by Małolepsza et al. [39]. The reaction mixture contained 0.5 cm^3^ of 0.05 M acetate buffer pH 5.6, 0.5 cm^3^ of 0.02 M guaiacol, 0.5 cm^3^ of 0.06 M H_2_O_2_ and 0.5 cm^3^ of supernatant. The absorbance was measured at 1-min intervals for 4 min at 480 nm. GPX activity was calculated using the absorbance coefficient 26.6 mM^−1^ cm^−1^. GPX activity was expressed as a change of peroxidase activity per fresh weight, expressed as U mg^−1^ fresh weight. Ascorbate peroxidase (APX, EC 1.11.1.11) activity was determined according to Nakano and Asada [40]. The reaction mixture contained 1.8 cm^3^ 0.1 M phosphate buffer pH 6.0, 0.02 cm^3^ of 5 mM sodium ascorbate, 0.1 cm^3^ of 1 mM H_2_O_2_ and 0.1 cm^3^ of the supernatant. Absorbance was monitored at a wavelength of 290 nm for 5 min, measured at 1 min intervals. APX activity was calculated using the absorbance coefficient of 2.8 mM^−1^ cm^−1^. Its activity was defined as the change of peroxidase activity per fresh weight, expressed as U mg^−1^ fresh weight.

### 2.4. Data Analysis

All physiological analyses were conducted in three biological replications (*n* = 3). One-way ANOVA was used to distinguish physiological reactions of different host plant tissues to the feeding of particular aphid species. Differences between means were determined using Tukey’s simultaneous test (HSD), and the level of significance was set at α = 0.05. Data are presented as means with standard deviation. The percentage change was calculated in the E_L_, TBARS and H_2_O_2_ content, and enzymatic activity in galled plant tissues relative to the intact leaves (as 100%). The obtained data were log-transformed. Differences in the mean percentage change in the content/activity of all analyzed parameters between tissues (undamaged, damaged, and gall) influenced by three aphid species were examined with factorial ANOVA/MANOVA, preceded by the Kolmogorov–Smirnov test. Means were separated by Tukey’s HSD test for unequal numbers, with the level of significance set at α = 0.05. All analyses were performed using Statistica 13.1 (StatSoft, Krakow, Poland) [41].

## 3. Results

Feeding of *C. compressa*, *E. ulmi* and *T. ulmi* affected H_2_O_2_ and TBARS level, electrolyte leakage from the cells, as well as antioxidant enzymes activity in host plant tissues. The percentage change in the level/activity of analyzed parameters was dependent on aphid species, tissue type as well as their interactions (Table 1).

### 3.1. Indicators of Oxidative Stress

The leaves with galls of *C. compressa* and *T. ulmi* showed a similar pattern of H_2_O_2_ concentrations. This molecule reached the highest level in tissues of undamaged parts of galled leaves, whereas its concentration in damaged parts was similar to intact leaves (Table 2). In *C. compressa* gall tissues, hydrogen peroxide content was 80% lower as compared to control leaves, while in *T. ulmi* galls, it was similar to the level in intact leaves (Figure 1C). On the other hand, H_2_O_2_ concentration in leaves with *E. ulmi* pseudogalls was higher in comparison with intact leaves, but differences were not significant (Table 2). 

Lipid peroxidation, as an indicator of oxidative stress, reflected the degree of leaf cell membrane damage due to galling. Undamaged parts of leaves galled by *C. compressa* were characterized by a 146% increase of TBARS, while only a 33% increase was noted in damaged parts of galled leaves in relation to control leaves (Figure 1A,B). The lowest content of TBARS was observed in gall tissues (Table 2). On the other hand, lipid peroxidation in the galls of *T. ulmi* reached more than a 5-fold increase as compared to the intact leaves (Figure 1C). TBARS content in undamaged parts of leaves galled by this aphid species was similar to control leaves, while it was approximately 90% higher in the damaged parts (Figure 1A,B). Pseudogalls of *E. ulmi* showed a clear decrease in TBARS level as compared to intact and galled leaves (Table 2). 

The feeding of all aphid species caused an increase in E_L_ in gall tissues in comparison with all other leave tissues (Table 2). The galling activity of *C. compressa* and *T. ulmi* caused a significant decrease in E_L_ levels in damaged parts of galled leaves (Figure 1B). In undamaged parts of all galled leaves, a downward trend of electrolyte leakage was also observed (Figure 1A).

### 3.2. Activity of Antioxidant Enzymes

The presence of *T. ulmi* resulted in a percentage decrease in CAT activity in galls (Figure 1C) and undamaged (Figure 1A) and damaged (Figure 1B) parts of galled leaves by 19.3%, 33.5% and 56.8%, respectively, compared to H_2_O_2_ concentration, cytoplasmic membrane condition, and changes in the activity of catalase and peroxidases, such as GPX and APX to intact leaves. Its lowest activity was observed in damaged parts of galled leaves. Gall formation by *C. compressa* induced a significant decrease in catalase activity only in damaged parts of galled leaves (Table 3). However, a slight increase of its activity in undamaged parts of galled leaves (Figure 1A) and in galls (Figure 1C) was also detected. On the other hand, *E. ulmi* feeding did not alter CAT activity significantly in host plant tissues (Table 3). 

The statistical analysis showed no significant differences in GPX activity in tissues under galling of *T. ulmi* (Table 3). The presence of other aphid galls altered GPX activity. The highest upregulation of that enzyme activity was recorded in undamaged parts of leaves galled by *C. compressa* and *E ulmi*. Its activity was almost 2-fold higher during *C. compressa* feeding as compared to the control rate (Figure 1A). In contrast, more than 60% lower GPX activity was detected in gall tissues of *C. compressa* compared to intact leaves (Figure 1C).

A significantly higher APX activity under the galling of all aphid species was observed. Aphid feeding in galls strongly increased APX activity in galled leaves and gall tissues (Table 3). The highest increase, almost 250% compared to intact leaves, was recorded in undamaged parts of galled leaves (Figure 1A). APX activity in galls was approximately 50–70% higher, depending on aphid species (Figure 1C). Significantly higher growth in that enzyme activity was observed in damaged parts of leaves galled by *C. compressa* than in damaged parts of leaves galled by *T. ulmi* (Figure 1B).

## 4. Discussion

The evolution of plants and insects has resulted in the development of strategies to avoid each other’s defenses [23]. However, many phytophagous species possess the ability to interfere with the tissues of their host plants to produce galls, which are often highly specialized structures [1,3]. Recent studies indicated that host plant physiological response to galling aphids is not unequivocal and depends on the insect species. 

Physiological and molecular reactions in plants against insect attacks are triggered by reactive oxygen species. Enhancement of ROS production was observed in numerous plant-aphid interactions [18,19,42,43]. The feeding activity of galling aphids can promote distinct structural and physiological changes, triggered by ROS alterations in host plant cells [12,15,44]. This study showed a marked increase of H_2_O_2_ under the galling of all analyzed aphid species, but only in undamaged parts of galled leaves. Surprisingly, a very low level of this molecule was measured in galls of *C. compressa*. Kot and Rubinowska [16] provided similar observations concerning different Cynipid species. On the other hand, the infestation of cottonwood by *Pempighus spyrothecae* Pass. [17], as well as Neotropical plants by galling Psyllidae and Cecidomyiidae [14,45] resulted in high H_2_O_2_ concentrations in the galls. The role of ROS in gall induction, development and functioning does not seem unequivocal. According to Morkunas and co-authors [18], ROS, as common intracellular and intercellular messengers with a broad spectrum of regulatory functions, are involved in many physiological processes. Hydrogen peroxide affects the activity of signaling compounds, such as MAP kinases, NADPH oxidase dependent on monomeric G protein, lipid-derived signals, Ca^2+^ channels, plant hormones, such as salicylic acid, jasmonic acid, abscisic acid, as well as ethylene and transcription factors, and thus they may also trigger gall morphogenesis [3,23]. However, a high concentration of H_2_O_2_ can be toxic to plants due to oxidative damage and may eventually lead to apoptotic cell death. On the other hand, ROS can cause oxidative damage to the insect midgut and reduce nutrient absorption [46]. 

The hydroxyl radicals may induce lipid peroxidation, which degrades cell structural components. In the current study, the highest amount of TBARS in galls and damaged parts of leaves galled by *T. ulmi* was observed as opposed to galling activity of *C. compressa* and *E. ulmi*, where the lowest amount of TBARS in galls was noted. However, higher TBARS levels were measured in different parts of leaves with true galls when compared to intact leaves. Previous reports indicated that the increase in lipid peroxidation was directly induced in Eucalyptus plants by a gall-forming psyllid [47] and in oak leaves by gall-inducing Cynipidae [16]. The products of lipid peroxidation can induce gene expression and are involved in plant signaling [48]. Some of them are highly reactive and participate in several physiological pathways, e.g., cell death, defense induction, signaling protein modification, or as secondary toxic cell messengers [49]. Both true and pseudo-galls wither after adult winged females emerge out of them [10]. It is possible that the cell death process was already triggered in *T. ulmi* galls similar to the Cecidomyiidae and Aspidosperma system, where signs of degradation and PCD in mature galls were found [13]. In turn, a low concentration of TBARS, accompanied by a decrease in H_2_O_2_ levels, suggested that the antioxidant defense system was activated in different parts of galled leaves and in the galls of *C. compressa* and *E. ulmi*, and was effective in free radical detoxification. Disorders in the integrity and stability of plasma membranes, measured by E_L_, are widely used as a test for stress-induced plant tissue damage. Electrolyte leakage triggered by all major stressors, including salinity, pathogen attack, heavy metals and wounding is usually accompanied by ROS accumulation and often results in PCD [50]. In our study, the highest level of E_L_ in gall tissues, H_2_O_2_ concentration, cytoplasmic membrane condition, and changes in the activity of catalase and peroxidases, such as GPX and APX were observed, which was similar to results in *Populus nigra* L.-*P. spyrothecae* system [17]. Aphids, as phloem feeders, penetrate plant cells and inject saliva, which plays a crucial role in preventing plant wound responses, but may also elicit plant reaction, resulting in damage during a later stage of infestation [18]. According to Demidchik and co-authors [50], electrolyte leakage from plant cells is mainly associated with increased K^+^ efflux through potassium channels. Potassium depletion in plants may enhance the activity of enzymes secondarily inducing PCD [43].

CAT, GPX and APX are enzymes representing the main enzymatic ROS scavenging mechanism in plants, and they are capable of scavenging H_2_O_2_ by various mechanisms, and APX is one of the central enzymes in this system [21]. The current research showed a significant increase in APX activity in all host tissues due to galling. Its highest activity was observed in undamaged parts of galled leaves in comparison to control. Various molecular forms of APX within the cells and organelles play important roles in developmental processes, including redox signaling [51]. Enhanced APX activity was also observed in winter triticale after *Sitobion avenae* (Fab.) and *Rhopalosiphum padi* L. aphid infestation [52] and soybean seedlings due to *Aphis craccivora* Koch feeding [25]. Ascorbate peroxidase activity generally increases under biotic and abiotic stresses along with other enzyme activities, such as CAT, SOD, and GSH reductase [53]. However, a significant increase in GPX activity was found in the present study only in undamaged parts of leaves galled by *C. compressa* and *E. ulmi*. In turn, the opposite reaction was observed in the galls of these aphid species. Gailite and co-authors [11] obtained similar results. The present study found no significant plant reaction caused by *T. ulmi* galling activity. Different responses of galls and galled leaves were also observed by Kot and Rubinowska [16] in the Cynipidae-oak system. Increased GPX activity is able to control the balance between H_2_O_2_ generation as a defense response and a decrease in hydrogen peroxide level to reduce oxidative damage in plant cells. Furthermore, reactive quinones and other oxidative radicals are produced by GPX in association with phenols, which act as feeding deterrents and generate toxins that reduce the digestibility of plant tissue [15,23,54]. It is possible that galling aphids manipulate the plant antioxidant system to avoid detrimental effects. Various CAT isoforms are involved in eliminating H_2_O_2_ generated during photorespiration, β-oxidation of fatty acids, and purine catabolism [55]. Catalase scavenges toxic and unstable ROS and directly converts them to oxygen and water, and in contrast to APX, it is more involved in H_2_O_2_ detoxification rather than regulation [51]. Our research showed a significant decrease in CAT activity in elm tissues due to *T. ulmi* feeding. A similar reaction was also found in damaged parts of leaves galled by *C. compressa*. On the other hand, slightly higher CAT activity was observed in the pseudogalls of *E*. *ulmi*, and undamaged parts of galled leaves, as well as in galls of *C. compressa*. The previous study documented a significant effect of *C. compressa*, *E. ulmi* and *T. ulmi* on photosynthesis photochemistry of different elm species. The strongest suppression of photosynthesis and pigment content due to *T. ulmi* feeding was found by Kmieć et al. [12]. A possible explanation of that phenomenon was the downregulation of CAT since an increase in its activity in leaves could protect chloroplasts, which presented sustained electron flows and were the main producers of ROS under stress conditions [56]. Mai and co-authors [42] have revealed that different CAT activities in plant responses to aphids suggest that plants have various mechanisms of aphid resistance. In turn, Shim et al. [57] indicated that CAT activity inhibition is a phenomenon that occurs in many plant species exposed to oxidative stress, and is related to salicylic acid accumulation. According to Apel and Hirt [58], when the balance of scavenging enzymes changes, compensatory mechanisms like APX and GPX upregulation and CAT downregulation, are induced in plants.

In conclusion, galling aphid feeding evoked various reactions of host plant tissues. Gall responses were usually quite different from those of leaves, although gall tissues had originated from leaf tissues as neoplastic formations. Physiological alterations of galls suggested manipulation that could be considered beneficial to aphids. Contrary to the hypothesis, different responses of H_2_O_2_, TBARS, CAT and GPX were found in true galls of *C. compressa* and *T. ulmi*. More detailed research is needed to clarify this phenomenon because gall induction and development is a remarkably dynamic process [1,15].

## Figures and Tables

**Figure 1 plants-11-00244-f001:**
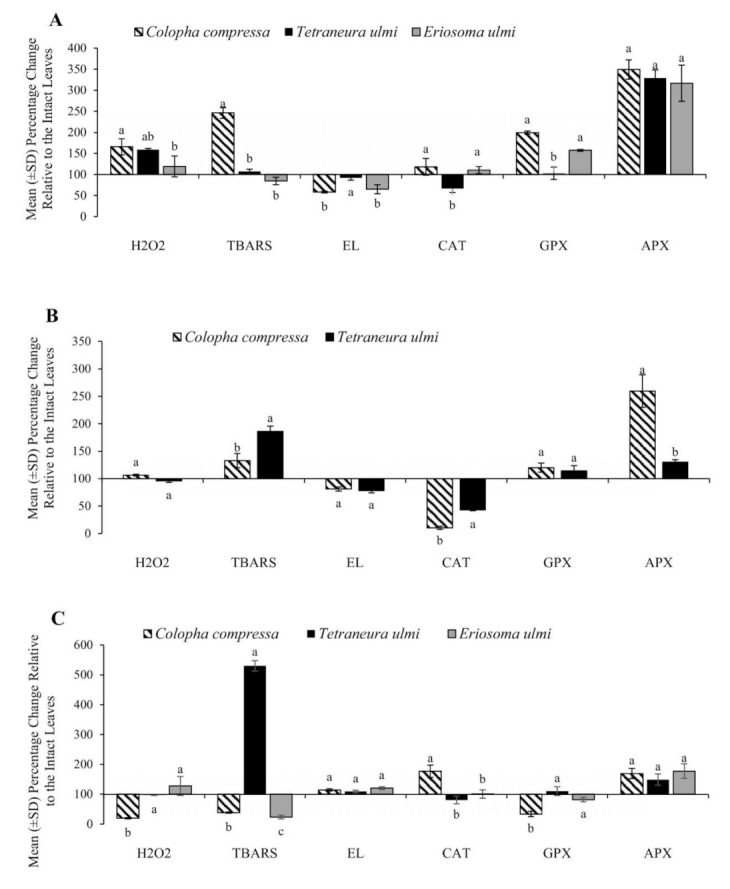
Mean (±SD) percentage change (relative to intact leaves as 100%) in level of hydrogen peroxide (H_2_O_2_), electrolyte leakage (E_L_) and thiobarbituric acid reactive substances (TBARS), as well as, ascorbate peroxidase (APX), peroxidase towards guaiacol (GPX) and catalase (CAT) activities in undamaged parts of galled leaves (**A**), damaged parts of galled leaves (**B**), and galls (**C**) of three aphid species (*Eriosoma ulmi* (L.), *Colopha compressa* (Koch) and *Tetraneura ulmi* (L.)). Bars sharing the same letter according to each parameter do not differ significantly at *p* ≤ 0.05 (Tukey’s test for unequal numbers).

**Table 1 plants-11-00244-t001:** Analysis of variance (ANOVA/MANOVA) in percentage changes relative to the intact leaves (as 100%) in hydrogen peroxide (H_2_O_2_) and thiobarbituric acid reactive substances (TBARS) content, electrolyte leakage (E_L_), as well as, catalase (CAT), peroxidase towards guaiacol (GPX) and ascorbate peroxidase (APX) activities with aphid species and host tissue as categorical factors.

Parameter	Aphid Species		Type of Host Tissue		Aphid Species x Type of Host Tissue	
	*df*	1	*df*	1	*df*	3
	*F*	*P*	*F*	*P*	*F*	*P*
H_2_O_2_	59.11	˂0.001	179.91	˂0.001	85.99	˂0.001
TBARS	171.52	˂0.001	92.17	˂0.001	267.23	˂0.001
E_L_	0.01	0.941 *	118.17	˂0.001	10.96	˂0.001
CAT	0.20	0.662 *	2.70	0.120 *	38.56	˂0.001
GPX	7.07	0.0171	171.37	˂0.001	55.82	˂0.001
APX	35.88	˂0.001	206.28	˂0.001	12.16	0.0002

An asterisk indicates no significance. Corresponding figure—Figure 1A–C.

**Table 2 plants-11-00244-t002:** The effect of *Eriosoma ulmi* (L.), *Colopha compressa* (Koch.), and *Tetraneura ulmi* (L.) galling on electrolyte leakage (E_L_), thiobarbituric acid reactive substances (TBARS) and hydrogen peroxide (H_2_O_2_) in tissues of *Ulmus glabra* Huds.

Host Plant/Aphid Species	Type of Tissue	E_L_ %	TBARS nmol g^−1^ FW	H_2_O_2_ nmol g^−1^ FW
*Eriosoma ulmi*	intact leaves	45.74 ± 1.1 ^b^	23.66 ± 1.3 ^a^	69.10 ± 8.2 ^a^
	galled leaves	29.74 ± 0.1 ^c^	20.13 ± 2.1 ^a^	80.53 ± 5.7 ^a^
	pseudogalls	55.09 ± 2.4 ^a^	5.59 ± 1.1 ^b^	85.26 ± 3.7 ^a^
*Colopha compressa*	intact leaves	30.22 ± 0.6 ^b^	16.19 ± 0.9 ^c^	49.62 ± 1.0 ^b^
	galled leaves UP	25.70 ± 0.7 ^c^	39.79 ± 1.7 ^a^	82.24 ± 8.3 ^a^
	galled leaves DP	24.50 ± 0.5 ^c^	21.26 ± 0.9 ^b^	52.78 ± 0.8 ^b^
	galls	34.60 ± 0.6 ^a^	6.19 ± 0.3 ^d^	9.67 ± 1.1 ^c^
*Tetraneura ulmi*	intact leaves	30.55 ± 0.6 ^ab^	22.81 ± 0.9 ^c^	38.81 ± 0.1 ^b^
	galled leaves UP	28.16 ± 1.3 ^b^	24.39 ± 0.3 ^c^	61.72 ± 1.3 ^a^
	galled leaves DP	23.82 ± 0.3 ^c^	42.50 ± 0.2 ^b^	37.24 ± 0.5 ^b^
	galls	33.34 ± 0.2 ^a^	122.15 ± 3.1 ^a^	37.85 ± 0.6 ^b^

Galled leaves UP—undamaged part of galled leaf lamina, galled leaves DP—damaged part of galled leaf lamina (with visible discoloration and/or corrugation), Mean (+SD) was calculated from three biological replicates for each treatment. Values with different letters, for each plant-aphid system, are significantly different at *p* ≤ 0.05 applying Tukey’s HSD test.

**Table 3 plants-11-00244-t003:** The effect of *Eriosoma ulmi* (L.), *Colopha compressa* (Koch.), and *Tetraneura ulmi* (L.) galling on antioxidant enzyme activities (catalase (CAT), peroxidase towards guaiacol (GPX), ascorbate peroxidase (APX)) in tissues of *Ulmus glabra* Huds.

Aphid Species	Type of Tissue	CAT U mg^−1^ FW	GPX U mg^−1^ FW	APX U mg^−1^ FW
*Eriosoma ulmi*	intact leaves	0.66 ± 0.0 ^a^	9.19 ± 0.1 ^b^	1.37 ± 0.1 ^c^
	galled leaves	0.73 ± 0.1 ^a^	14.47 ± 0.3 ^a^	4.31 ± 0.2 ^a^
	pseudogalls	0.67 ± 0.1 ^a^	7.53 ± 0.3 ^c^	2.41 ± 0.1 ^b^
*Colopha compressa*	intact leaves	0.36 ± 0.1 ^a^	6.89 ± 0.3 ^b^	0.95 ± 0.02 ^d^
	galled leaves UP	0.41 ± 0.1 ^a^	13.75 ± 0.8 ^a^	3.30 ± 0.2 ^a^
	galled leaves DP	0.04 ± 0.03 ^b^	8.26 ± 0.2 ^b^	2.46 ± 0.3 ^b^
	galls	0.49 ± 0.2 ^a^	2.34 ± 0.7 ^c^	1.60 ± 0.1 ^c^
*Tetraneura ulmi*	intact leaves	1.37 ± 0.2 ^a^	13.18 ± 0.1 ^a^	1.86 ± 0.1 ^c^
	galled leaves UP	0.90 ± 0.1 ^bc^	13.45 ± 0.8 ^a^	6.09 ± 0.1 ^a^
	galled leaves DP	0.57 ± 0.02 ^c^	14.97 ± 0.1 ^a^	2.41 ± 0.04 ^b^
	galls	0.96 ± 0.04 ^b^	14.45 ± 0.7 ^a^	2.76 ± 0.3 ^b^

Galled leaves UP—undamaged part of galled leaf lamina, galled leaves DP—damaged part of galled leaf lamina (with visible discoloration and/or corrugation), Mean (+SD) was calculated from three biological replicates for each treatment. Values with different letters, for each plant-aphid system, are significantly different at *p* ≤ 0.05 applying Tukey’s HSD test.

## Data Availability

Not applicable.

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
