# Peer review of "The Variation of Selected Physiological Parameters in Elm Leaves (Ulmus glabra Huds.) Infested by Gall Inducing Aphids"

_plants, 2022, doi:10.3390/plants11030244_

Round 1

Reviewer 1 Report

The paper submitted by Kmiec and colleagues deals about the gall formation induced by three aphid species (Eriosoma ulmi, Colopha compressa and Tetraneura ulmi) comparing their effects on Ulmus glabra tissues. Authors reported, in particular, a marked increase of H2O2 under galling of all the aphid species, but only in undamaged parts of galled leaves.

The introduction is clearly written and properly structured and no inappropriate self-citations by Authors is present. The experimental plan has been properly defined and methods adequately reported. Results are properly reported and discussed, but I think that a graphical abstract could help readers to better understand them. Indeed, the observed results are different not only comparing aphid species, but also in respect to the previous hypotheses about gall induction effects.  The presence of a figure summarizing the results (at the end of results) could be of great importance for readers considering that gall induction and development is a remarkably dynamic and complex process.

As a whole I think that the manuscript is properly structured and written, methods and data are adequately reported and conclusions are supported by data so that I suggest to accept the manuscript in the present form with thee introduction of a new figure summarizing results.

Author Response

Response to Reviewer’s comments

The authors would like to thank the Reviewer for careful review of our manuscript and providing us with their comments and suggestion to improve the quality of the manuscript. Our response follows.

Author’s Reply to the Report of Reviewer 1

  1. Comment: “Indeed, the observed results are different not only comparing aphid species, but also in respect to the previous hypotheses about gall induction effects. The presence of a figure summarizing the results (at the end of results) could be of great importance for readers considering that gall induction and development is a remarkably dynamic and complex process.“

Response: Done. We attached the summary of our results as a graphical abstract in separate file, not attached in main text.

Reviewer 2 Report

I do not have major issues with the manuscript by Kmiec et al. A few point are as follows:

  • line 31: delete `most`
  • lines 82-83: sentence incomplete, please rewrite
  • line 86: delete `in different parts`
  • line 101: replace `galled leaves` with `galls`
  • line 258: I suggest to use `evolution` instead of `coevolution`, because it is more precise and less speculative.
  • lines 304-307: rewrite sentence, as it is difficult to understand
  • line 322: correct to `Aphis craccivora`

Author Response

Response to Reviewer’s comments

The authors would like to thank the Reviewer for careful review of our manuscript and providing us with their comments and suggestion to improve the quality of the manuscript. Our response follows.

Author’s Reply to the Report of Reviewer 2

Comments: 

  • line 31: delete `most`
  • lines 82-83: sentence incomplete, please rewrite
  • line 86: delete `in different parts`
  • line 101: replace `galled leaves` with `galls`
  • line 258: I suggest to use `evolution` instead of `coevolution`, because it is more precise and less speculative.
  • lines 304-307: rewrite sentence, as it is difficult to understand
  • line 322: correct to `Aphis craccivora`

Response: Done. All the Reviewer's suggestions were taken into account and included in the manuscript.